# Nitric Oxide Regulates Seed Germination by Integrating Multiple Signalling Pathways

**DOI:** 10.3390/ijms24109052

**Published:** 2023-05-21

**Authors:** Yue Zhang, Ruirui Wang, Xiaodong Wang, Caihong Zhao, Hailong Shen, Ling Yang

**Affiliations:** 1State Key Laboratory of Tree Genetics and Breeding, School of Forestry, Northeast Forestry University, Harbin 150040, China; zhangyue19970309@163.com (Y.Z.);; 2Research Center of Korean Pine Engineering and Technology, National Forestry and Grassland Administration, Harbin 150040, China

**Keywords:** nitric oxide, seed dormancy, seed germination, plant hormones, abiotic stress

## Abstract

Seed germination is of great significance for plant development and crop yield. Recently, nitric oxide (NO) has been shown to not only serve as an important nitrogen source during seed development but also to participate in a variety of stress responses in plants to high salt, drought, and high temperature. In addition, NO can affect the process of seed germination by integrating multiple signaling pathways. However, due to the instability of NO gas activity, the network mechanism for its fine regulation of seed germination remains unclear. Therefore, this review aims to summarize the complex anabolic processes of NO in plants, to analyze the interaction mechanisms between NO-triggered signaling pathways and different plant hormones such as abscisic acid (ABA) and gibberellic acid (GA), ethylene (ET) and reactive oxygen species (ROS) signaling molecules, and to discuss the physiological responses and molecular mechanisms of seeds during the involvement of NO in abiotic stress, so as to provide a reference for solving the problems of seed dormancy release and improving plant stress tolerance.

## 1. Introduction

Plants reproduce by producing seeds, and the seed dormancy-germination process, which initiates the plant life cycle, is crucial for the establishment of seedlings and their ability to survive adverse external environments [1,2]. Seed dormancy refers to the physiological process of primary dormancy, which prevents the germination of viable seeds under favorable conditions [3,4,5]. The main types of dormancy include physiological dormancy, morphological dormancy, physical dormancy, and combinational dormancy [6,7]. Physiological dormancy occurs mainly in the seeds of gymnosperms and angiosperms and has been extensively studied in model plants such as *Arabidopsis thaliana* [8] and *Nicotiana tabacum* [9]; temperature stratification can break this type of dormancy. Morphologically dormant seeds are very clearly differentiated morphologically, but the internal embryo is not fully developed, which leads to delayed germination. Physical dormancy is mainly caused by mechanical hindrance from the external seed coat, and damage to the seed coat can induce seed germination under unfavorable conditions and break seed dormancy. Most seeds have a variable integrated dormancy state, which may have both physiological and morphological dormancy or both physiological and physical dormancy [6,7]. However, the combined dormant state of seeds can change depending on environmental conditions, and seeds can initiate germination to break dormancy when the external environment becomes favorable for plant growth [10]. There are three main stages in the process of breaking seed dormancy and germination: Initially, dry seeds absorb water (imbibition). Then, the embryo begins to expand and enzymes become activated. Finally, the protrusion of the embryo radicle and cotyl elongation complete the process of germination [4,11]. Thus, the transition from seed dormancy to germination is a complex process. In addition, this process is affected by internal genetic and external environmental factors as well as available nitrogen sources [12]. Light, temperature, nitrogenous compounds as signal inputs for seed dormancy and germination, activate changes in the levels of internal hormones such as ABA and GA. Seed dormancy is mainly induced by ABA. However, Seed germination is mainly induced by GA. The dynamic balance between ABA and GA is an important factor for breaking dormancy and promoting germination [12].

Nitrogen is an essential nutrient for plant growth and development. Reactive nitrogen species (RNS) have important regulatory functions in plant physiological activities. Recent studies have reported that the main forms of RNS in plants include nitric oxide (NO), nitrogen dioxide (NO_2_), peroxynitrite (ONOO^−^), nitrate (NO_3_^−^), and nitrite (HNO_2_) [13,14,15,16]. Among these forms, NO acts as a key gaseous molecule for regulating seed germination and improving crop productivity. It can promote seed germination by regulating ABA metabolism and GA synthesis pathways [11].

Evidence is accumulating that NO donors, such as sodium nitroprusside (SNP), S-nitroso-N-acetyl-D-penicillamine (SNAP), and S-nitrosoglutathione (GSNO), can break seed dormancy. In contrast, the NO scavenger (2-(4-carboxyphenyl)-4, 4, 5, 5-tetramethylimidazoline-1-oxyl-3 oxide; cPTIO) inhibited seed germination [17]. For example, Indian mustard (*Brassica juncea* L.) was treated with 0–250 μM SNP, and it was found that 100 μM SNP was the best promoter of seed germination [18]. A low SNP concentration of 25 μM reduced dormancy of *Arabidopsis* seeds, while an SNP concentration of more than 250 μM inhibited germination [19]. Fumigation of apple (*Malus domestica* Borkh.) embryos with 3 mM SNAP promoted germination, while 0.3 mM cPTIO treatment significantly inhibited germination [20]. SNAP treatment of 500 µM significantly accelerated the germination rate of chickpea seeds [21]. The germination index of rice seed treated with 100 μM GSNO significantly increased [22].

In addition, during seed germination of different types of plants, NO signals can affect physiological responses to a variety of factors, including light, temperature, drought, and salinity [12,23,24]. Moreover, a large number of studies have emphasized that the interaction between NO and phytohormones is crucial in seed biology [25]. Therefore, this review aims to summarize the interactions of NO with a variety of pathways to enhance our understanding of the physiological and potential molecular mechanisms regulating seed germination.

## 2. Main Text

### 2.1. Synthesis and Decomposition of Nitric Oxide in Plants

Nitrogen displays a range of oxidation states between a strongly oxidized form and a fully reduced form and can produce NO through oxidation or reduction mechanisms [26]. Current studies have shown that there are two pathways for plant NO synthesis (Figure 1). One is through a nitrate/nitrite-dependent reduction pathway, and the other is through an NO synthase (NOS)-mediated oxidation pathway [26,27,28,29].

The reduction pathway can be catalyzed by a variety of reductases, including nitrate reductase (NR), NO-forming nitrite reductase (NOFNiR), or a non-enzymatic pathway that occurs in the mitochondrial electron transport chain (mETC). Among these pathways, the mETC-dependent reductions of nitrite to NO are mainly through electron transfer between complex III and complex IV [27,28,30,31]. A recent study reported that there are two genes encoding NR in the *Arabidopsis* genome, NIA1/NR1 and NIA2/NR2 [31]. NR-mediated reduction of nitrite to NO is the major source of NO in higher plants, and this reaction occurs under aerobic or acidic conditions when concentrations of nitrite and nitrate are high [27,32]. With NR, nitrate is first reduced to nitrite, and then nitrite is reduced to NO [27].

The oxidative pathway is linked to the oxidation of L-arginine (Arg). Similar to NO synthesis in mammals, plants catalyze the oxidation of Arg to citrulline and NO using NOS [33,34]. Additionally, there is a polyamine synthesis pathway with Arg as a precursor, and studies have reported that increased levels of polyamines (PAs), such as spermine (Spm) and spermidine (Spd), lead to NO release in a variety of plants [35].

The non-enzymatic pathway to produce NO is utilized under anaerobic conditions with nitrite as the electron acceptor, which maintains plant respiration [27]. Under hypoxic conditions, plant cells produce NO mainly through the mETC [36], suggesting that plants may induce NO production through a variety of pathways (Figure 1). Therefore, future research on the complex mechanism of NO production is needed.

NO is unstable, and when it reacts with oxygen, NO_3_^−^ or NO_2_^−^ is formed [26]. NO can be scavenged both by reacting with ROS and by interacting with superoxide anions (O_2_^−^) to produce ONOO^−^ [37,38]. It can also react with glutathione (GSH) to form GSNO, which is further reduced to glutathione disulfide (GSSG) and ammonia (NH_3_) by GSNO reductase (GSNOR) [39,40]. In addition, NO removal can be achieved by oxidation of NO to nitrate by hemoglobin (Figure 1). Maintaining the homeostasis of NO in plant cells by controlling its concentration is a significant factor for the seed germination process [41].

### 2.2. Nitric Oxide and Phytochrome Signaling Pathways Jointly Regulate Seed Germination under Light

Light is an important signal that affects plant survival; through photosynthesis, plants obtain energy to maintain growth. Current studies have reported that plants activate multiple photoreceptors, which include blue light receptors, red light/far-red light receptors and other photoreceptors that respond to light signals at different wavelengths [42,43]. There are five phytochrome (PHY) proteins in *Arabidopsis*; PHYA and PHYB are the photoreceptors primarily responsible for light-induced seed germination. PHYA responds to far-red light, and PHYB responds to red light [44,45].

Beligni explored the effect of NO on plant photomorphogenesis for the first time, revealing that NO donor SNP can induce lettuce seed germination depending on light [46]. Recent studies have shown that empress trees (*Paulownia elongata*) have physiological dormancy characteristics, and the seed germination rate after SNP treatment and continuous illumination for 12 h is significantly higher than that of unilluminated seeds [47]. NO stimulation also led to greening of etiolated seedlings of wheat (*Triticum aestivum* L. cv. Buck Patacon), which contained 30–40% more chlorophyll than controls when treated with 100 μM SNP and grown in the dark [46]. Another photoresponse to NO stimulation is the inhibition of hypocotyl and internode elongation. The germination rate of lettuce (*Lactuca sativa* L. cv. Grand Rapids) seeds treated with 100 μM SNP or 100 μM SNAP (both NO donors) and given red light (20 μmol s^−1^m^−2^) pulses was greater than 90%. For *Arabidopsis* (*Arabidopsis thaliana* (L.) Heynh.) treated with NO donors under dark conditions or potato (*Solanum tuberosum* (L.) cv. Pampeana) tubers under low light intensity, hypocotyl elongation was significantly inhibited, and the internode length of the plants was shortened [46]. These results indicate that NO-dependent photoreceptors affect seed germination, hypocotyl elongation, and plant greening.

In addition, the mechanism of action in *Arabidopsis* in which NO interacts with PHY to collectively affect seed germination was further elucidated (Figure 2). Exogenous application of the nitrogenous compound potassium nitrate (KNO_3_) to *Arabidopsis* seeds, followed by exposure to far-red light/red light for a short period of time, stimulates the activity of PHYA and PHYB, which in turn will promote rapid seed germination [48]. Within this process, PHYB is particularly crucial during its early stages and regulates seed germination by controlling the stability of the protein PHYTOCHROME-INTERACTION FACTOR 1 (PIF1) under red/far-red light [44]. An antagonistic relationship between PHYB and PIF1 has been reported (Figure 2). In the presence of red light, PHYB moves from the cytoplasm to the nucleus and promotes seed germination promoting expression of the 26S proteasome pathway leading to PIF1 degradation [49]. Significant hypocotyl elongation and increased anthocyanin levels were found in the NO biosynthesis-deficient *Arabidopsis nia1, 2noa1-2* mutant after exposure to red light pulses. Moreover, PHYB protein content was reduced and *PIF1*, *PIF3*, and *PIF4* expression was enhanced in the red light-induced mutants, and these results further support that NO and PHYB interact during seed germination [50]. In addition, NO in light can increase the accumulation of PHYB, which mediates the degradation of PIF3, leading to the inhibition of initial root elongation in *Arabidopsis* [51]. *SOMNUS* (*SOM*), as a direct target gene downstream of PIF1, can affect the expression levels of related genes in the abscisic acid (ABA) and gibberellic acid (GA) signaling pathways in photochrome-dependent photomorphogenesis (Figure 2). In the ABA signaling pathway, SOM promotes ABA biosynthesis by activating three ABA anabolic genes, ABA-DEFICIENT1 (*ABA1*), NINE-CIS-EPOXYCAROTENOID DEOXYGENASE6 (*NCED6*), and *NCED9*, and inhibits the expression of an ABA catabolic gene (*CYP707A2*), leading to inhibition of seed germination. In the GA signaling pathway, SOM suppresses GA biosynthesis by inhibiting GA synthesis metabolism genes (*GA3ox1* and *GA3ox2*) and activating a catabolic gene (*GA2ox2*), leading to lower GA and inhibition of seed germination (Figure 2) [52]. In addition, the DELLA protein plays an important regulatory role in the GA signaling pathway by affecting the activity of PIF transcription factors under red light irradiation [50]. An antagonistic relationship between NO and GA was found during seed germination under red light irradiation. With increased concentration of the nitric oxide donor SNP, GA accumulation was inhibited, and GA-regulated DELLA protein accumulation was induced, which reduced the expression of *PIFs*, resulting in the inhibition of hypocotyl elongation (Figure 2) [50].

In addition to the regulation of seed germination under light through PHYB and its interacting protein PIFs, a member of the basic helix-loop-helix (bHLH) transcription factor family, LONG HYPOCOTYL IN FAR-RED (HFR1), was found to form dimers with PIF1, reducing the transcriptional activity of PIF1 (Figure 2). HFR1 accumulation increased under light stimulation, and the interaction of HFR1-PIF1 produced more precise regulation of light-initiated seed germination [53]. Recently, NO accumulation under red light conditions was found to not only inhibit PIF1 transcription but also enhance the HFR1-PIF1 interaction, ultimately leading to a weakened inhibitory effect of PIF1 on seed germination (Figure 2) [54].

NO regulation of seed germination is also associated with blue photoreceptors and functions in cooperation with regulatory factors such as methyl jasmonate (MJ) and phospholipase D (PLD). Blue light at 20 μmol m^−2^ s^−1^ inhibited germination of dormant wheat; however, when NO and MJ were applied simultaneously, the inhibitory effect of blue light was attenuated. Inhibition of synthesis of the ABA coding gene *TaNCED1* and increased expression of the metabolic gene *TaABA8′OH-1* decreased the ABA content, thereby reducing grain dormancy [55]. NO was also involved in the specific response of tomato seeds to blue light under osmotic stress. Exogenous addition of 0.2 mM S-nitrosoglutathione (SNG), an NO donor, promoted seed germination under 10 μmol m^−2^ s^−1^ blue light, while exogenous addition of 0.1 mM PTIO, an NO scavenger, inhibited seed germination [56]. PLD, a member of an important family of phospholipases in plants (Figure 2), hydrolyzes phospholipids to produce phosphatidic acid (PA) [57]. Activation of PLD and the PA produced by its hydrolysis during seed germination can regulate cytoskeletal organization and mediate the transport of Ca^2+^ signals in the cytoplasm [57,58]. During seed germination, NO content and PLD activity are increased by light. Indeed, in lettuce seeds, light induces NO production, stimulates PLD activity, and ultimately leads to the production of more PA, which in turn promotes seed germination [59]. However, further studies are still needed on the specific functions of light-induced NO production and the type of PA produced by PLD during seed germination.

### 2.3. The Crosstalk between Nitric Oxide and Plant Hormone Signaling Pathways in Seed Dormancy and Germination

#### 2.3.1. Nitric Oxide and Abscisic Acid Signaling Pathways Jointly Regulate Seed Germination

It is well known that the phytohormone ABA often plays a key role in the induction of seed dormancy [6,60]. In contrast, the presence of NO has been reported in many studies to reduce the sensitivity of seeds to ABA, and the interaction of NO and ABA in the regulation of seed germination has been demonstrated in plants such as *Arabidopsis*, switchgrass and warm-season C4 grasses [61,62,63,64]. The seed coat of *Arabidopsis* is composed of a dead coat and a layer of living aleurone cells. As the only endosperm tissue, the living, single-cell thick, aleurone layer stores lipids and proteins during seed maturation [65,66]. Therefore, the aleurone layer is the single most important factor determining seed dormancy. *Arabidopsis* and barley maintain the growth of embryos by storing lipids and proteins in the aleurone layer and increasing the expression of the NO oxide synthase gene (*AtNOS1*), while NO acts upstream of ABA and GA to increase biosynthesis of GA and inhibit biosynthesis of ABA, which conserves nutrients and promotes the normal germination of seeds [67]. The enzyme NCED catalyzes ABA biosynthesis to produce xanthoxin (Figure 3). Then, xanthoxin is converted to abscisic aldehyde by short-chain dehydrogenase reductase (SDR1), followed by oxidation of abscisic aldehyde oxidase (AAO3) to ABA catalyzed by ABA3 (Figure 3). Further, 8′-hydroxylation by the enzyme CYP707A, which can hydrolyze ABA into phaseic acid (PA) and dihydrophaseic acid (DPA), is thought to be the main pathway of ABA catabolism [68]. In the model plant *Arabidopsis*, NO induced rapid accumulation of ABA in the endosperm layer during early germination, leading to reduced ABA concentration in the embryo and thus promoting seed germination by upregulating the expression of ABA 8′-hydroxylase encoded by CYP707A2 and inducing catabolism of ABA [61,69]. In model plants, it is well established that NO regulates seed germination by affecting the expression of genes associated with ABA metabolism. A recent study reported that NO produced by NOS or NR in potato tubers significantly induced sprouting of the tubers by promoting the expression of the ABA catabolic gene *StCYP707A1* while inhibiting the expression of the ABA biosynthesis-related gene *StNCED1*, thereby reducing ABA content and altering the ABA-GA balance [70]. Additionally, in the embryonic axes isolated from dormant apple embryos, NO reduced ABA concentration by promoting the expression of ABA catabolism gene *CYP707A2* and inhibiting the expression of ABA synthesis genes (*NCED3* and *NCED9*) [71].

NO also affects ABA signaling in guard cells during seed dormancy [72]. The mechanism is shown in Figure 3: ABA binds to its receptor PYR/PYL/RCAR, resulting in the inactivation of type 2C protein phosphatases (PP2C), which stimulates the action of SNF1-related protein kinase 2 (SnRK2). The transcription of ABSCISIC ACID INSENSITIVE5 (ABI5), an important repressor of seed germination, is promoted by phosphorylation, which leads to seed dormancy. When exogenous SNP is added or endogenous NO accumulates, NO-dependent protein modifications repress ABA signaling through S-nitrosylation of two important kinases of the SnRK2 family, SnRK2.2 and SnRK2.3, which in turn break seed dormancy [73,74,75]. In addition to mediating S-nitrosylation of SnRK2s, crosstalk between NO and ABA signaling also plays an important role in seed germination by indirectly regulating the expression of ABI5 in the N-end rule pathway (Figure 3). Group VII of the ethylene response factor (ERF/AP2) family (ERFVIIs) has a conserved the N-terminal domain. In the presence of nitric oxide, ERFVII is destabilized and exposed N-terminal Cysteine (Cys) residues are susceptible to oxidation, followed by arginylation by Arg-tRNA protein transferase and finally ubiquitination by E3 ligases, leading to repression of the downstream transcription factor ABI5, which in turn promotes seed germination [76,77].

#### 2.3.2. Nitric Oxide and Auxin Signaling Pathways Jointly Regulate Seed Germination

During seed germination in higher plants, NO-mediated release of embryonic dormancy is positively correlated with ethylene synthesis [68]. Ethylene synthesis begins with the activation of methionine (Met) by S-adenosyl-L-methionine synthetase (SAM synthetase) to produce S-adenosyl-methionine (S-AdoMet) (Figure 4). The transformation to 1-aminocyclopropane-1-carboxylic acid (ACC) and the by-product 5′-methylthioadenosine (MTA) is catalyzed by ACC synthase, S-adenosyl-L-methionine methylthioadenosine-lyase (ACS), and then ACC is oxidized using ACC oxidase (ACO) to produce ethylene, CO_2_, and hydrogen cyanide (HCN) [78]. MTA regenerates Met through the methionine cycle [78]. To date, five ethylene receptors have been identified in the model plant *Arabidopsis*. Among them, ethylene resistant 1 (ETR1) plays an important role in NO-mediated ethylene signal transduction [79,80]. Ethylene signaling is regulated by constitutive triple response 1 (CTR1), and when the ethylene receptor ETR1 on the endoplasmic reticulum receives the ethylene signal, it leads to the inactivation of the receptor and CTR1. CTR1 is a negative regulator of ETHYLENE INSENSITIVE2 (EIN2), while EIN2 acts as a positive regulator of ethylene signaling by increasing the activity of the nuclear transcription factor EIN3, which in turn activates the transcription of downstream ethylene response elements such as ethylene-responsive factor (ERF) and other genes that promote seed germination [81]. Recent studies have also shown that EIN2, a key factor in the ethylene signaling pathway, is involved in jasmonic acid (JA)-induced primary root growth through effects on NO accumulation [82]. The involvement of ETR1, EIN2, and EIN3 in NIA1/2-mediated NO production is associated with salicylic acid (SA)-induced stomatal closure. All these findings confirmed the important connections between NO and the ethylene signaling pathway [83].

NO-ethylene interactions promote seed germination in a variety of plants. For instance, exogenous NO donors SNP or SNAP can increase the germination of apple embryos, while the ethylene synthesis inhibitor aminooxyacetate acid (AOA) represses the germination of embryos by NO donors [84]. Moreover, the effect of NO on ethylene production was associated with the accumulation of ROS. Pretreatment of apple embryos with HCN or SNP leads to short-term accumulation of ROS (H_2_O_2_ and O_2_^−^), which breaks the deep dormancy of apple embryos. However, the exogenous addition of ethylene glycol or the ethylene precursor ACC induced normal growth of embryonic roots and greening of cotyledons after embryo germination [85]. Gniazdowska et al. investigated the relationship between ethylene and the role of NO and HCN in mediating seed germination and demonstrated that early in the process of seed germination, NO and HCN pretreatment could directly change the activities of ACS and ACO, key enzymes of the ethylene synthesis process. Moreover, ACC can be oxidized to ethylene by ROS free radicals in the early stage of seed germination, leading to increased ethylene content [86]. In addition, a recent study found that increased SNP during germination of *Sorbus pohuashanensis* is related to ethylene synthesis and ROS accumulation. NO promoted hypocotyl and radicle growth by inducing ethylene biosynthesis and ROS accumulation, enhancing the capacity of antioxidant defense systems, and reducing ABA content [87]. Exogenous NO donors increase the concentrations of endogenous ethylene and ROS, promoting seed germination. Studies have shown that exogenous ethylene treatments can increase the production of ROS in the embryo axis and break sunflower seed dormancy. Moreover, exogenous application of 0.1 mM methyl viologen (MV), an ROS-inducing herbicide, can alleviate the inhibitory effect of ABA on sunflower seed germination and reduce the ABA content in the embryo axis, reducing the inhibitory effect on seed germination [88].

Karrikin-1 (KAR1) is an active compound derived from smoke that affects seed germination in different types of plants [89]. Sami et al. showed that *Brassica oleracea* L. (Chinese cabbage) seeds treated with NO and KAR1 had reduced ABA content and increased GA and ROS content, but the activities of antioxidant system enzymes, including catalase (CAT) and glutathione reductase (GR), increased [90]. The expression levels of ethylene biosynthesis genes (*BOACS7*, *BOACS9*, and *BOACS11*), ethylene receptors (*BOETR1* and *BOETR2*), and an ACC oxidase ACO gene (*BOACO1*) were significantly increased with induction by NO and KAR1 [90]. Furthermore, KAR1 and HCN treatments reduced the ABA content and increased the GA content in *B. oleracea* seeds with secondary dormancy and induced the expression of ethylene synthesis genes and ethylene receptor genes, releasing seed dormancy [91]. However, the mechanism of the interaction of NO with KAR1 and HCN to release seed dormancy by mediating ethylene synthesis still requires further exploration. Further, ethylene, HCN, and NO gases applied directly to dry seeds can break seed dormancy [92]. Therefore, in addition to the application of solutions, the direct application of gas can induce release from seed dormancy; future research can reveal new insights on the process of seed dormancy release in different types of plants.

#### 2.3.3. Nitric Oxide and Gibberellin Signaling Pathways Jointly Regulate Seed Germination

NO-GA interactions often are antagonistic to the regulatory role of ABA during seed germination [25]. Exogenous SNP and GA_3_ can improve seed viability and seed germination in grapevines [93]. In *Arabidopsis*, NO increases GA biosynthesis by inducing downstream transcript levels of gibberellin 3-oxidase 1 (*GA3ox1*) and *GA3ox2*, which promotes seed germination [67]. NO not only regulates the GA signal but also interacts with ethylene to regulate seed germination. For example, the exogenous hormone ethephon or ethylene precursor ACC can release ethylene and GA_3_ to break seed dormancy and induce seed germination [94,95]. Additionally, seeds are sensitive to dry environments, and the physiological response of seeds to GA_3_ and ethylene is enhanced with dry storage and stratified environments [95]. Kepczynski et al. further investigated the interactions between NO, GA_3_, and ethylene during seed germination, and confirmed that the stimulation of seed germination by NO and GA_3_ was positively correlated with ethylene production by measuring the ethylene content of the radicle before it protruded from the seed coat. The exogenous addition of the NO inhibitor cPTIO or the ethylene receptor inhibitor 2,5-norbornadiene (NBD) inhibited the induction of seed germination by NO and GA_3_, whereas the repression was alleviated when ethylene was added in combination with NBD [96]. These results suggest that the release of seed dormancy by GA_3_ and NO is dependent on the function of ethylene.

#### 2.3.4. Nitric Oxide Regulates Seed Germination Together with Polyamines and Jasmonic Acid Signaling Pathways

In addition to inducing the synthesis of GAs to promote seed germination, NO can stimulate seed dormancy release by regulating the synthesis of JA in wheat grains and apple embryos [55,71]. It was found that NO could induce the expression of the genes *AOS1*, *JMT*, and *JAR1* encoding JA and JA derivatives JA methylester (MeJA) and JA-isoleucine (JA-Ile), respectively, promoting seed germination [71].

Polyamines (PAs) are an important plant growth regulator [97]. Studies have shown that PA can rapidly induce the release of NO in embryogenic suspension cultures of *Araucaria angustifolia* [98]. In plant cells, the synthesis of PA is related to the ethylene synthesis pathway (Figure 4). SAM is central to ethylene synthesis, then decarboxylated SAM (dcSAM) is produced under the action of SAM decarboxylase (SAMDC), and finally, PA is produced by PA synthase. Moreover, PA oxidase (PAO) catalyzes the oxidative deamination of PAs, producing H_2_O_2_ [99,100,101]. However, increased levels of PAs, including (Spm) and Spermidine (Spd) also resulted in a significant increase in NO released by the Arg-dependent pathway. [101]. These results suggest a close regulatory network between NO and PAs, but the mechanism of their interaction during the transition from seed dormancy to germination is not well understood. Recent studies have reported that at the transcriptional level, NO can increase the expression of PA biosynthetic genes *MdPAO*, *MdSAMDC1*, and *MdSPDS2a* to release apple embryonic dormancy and induce seed germination [101,102].

### 2.4. Regulation of Nitric Oxide on Seed Germination under Abiotic Stress

#### 2.4.1. Mechanism of Nitric Oxide Action on Seed Germination under Salt Stress

The NO synthetic protein is AtNOA1 in *Arabidopsis* genome and OsNOA1 in *Oryza sativa* L. genome. The molecular mechanisms of the two proteins are similar in regulating the salt tolerance of plants during seed germination. *OsNOA1* can complement the *atnoa1* mutant phenotype and the loss of chlorophyll synthesis by promoting NO release, and improve the salt tolerance of plants during seed germination by decreasing the Na^+^/K^+^ ratio in the mutant (Table 1) [103,104]. Seed germination can also be promoted by magnetic fields. For example, NO promoted α-amylase activity and the physiological response of seed germination in maize (*Zea mays*) under magnetic initiation [105]. In soybean (*Glycine max* (L.) Merrill) seeds, *GmNOS2* and *GmNR1* may be ideal candidate genes involved in NO production (Table 1). Magnetopriming-mediated NO production decreased the Na^+^/K^+^ ratio and maintained the balance of ABA, GA, and ondole-3-acetic acid (IAA) hormones, thereby improving the salt tolerance index of soybean [106]. NO not only affects ion homeostasis under salt stress but also alleviates oxidative damage in plants by upregulating the antioxidant defense system and maintaining ROS homeostasis under stress conditions [107,108,109,110]. In tomatoes, overexpression of tomato glutathione reductase (*SlGR*) led to the accumulation of more antioxidant substances under 100 mM NaCl treatment (Table 1) [111]. Additionally, 100 μM SNP, by releasing cyanide, reduced lipid peroxidation and increased the activity of antioxidant enzymes, which in turn increased the speed and rate of germination [112]. Recently, Hajihashemi et al. found that RNS and ROS, together with Ca^2+^, regulate the physiological response of *Chenopodium quinoa* seeds under salt stress. Treatment with a combination of 5 mM CaCl_2_, 5 mM H_2_O_2_, and 0.2 mM SNP leading to starch hydrolysis and increased water-soluble sugar content, effectively alleviating the adverse effects of salt stress on seed germination [113].

In addition to promoting seed germination under salt stress by providing NO through SNP, KNO_3_, the major form of nitrate, is useful for breaking seed dormancy in plants such as *Arabidopsis* [114] and maize [115]. However, the specific mechanism of nitrate-mediated responses of plants to salt remains unclear. Studies have reported that the main regulators of nitrate signaling include Chlorate-resistant1 (NRT1.1, CHL1, NPF6.3), Arabidopsis nitrate regulated 1(ANR1), Teosinte branched1/cycloidea/proliferating cell factor1-20 (TCP20), and NIN-like protein (NLP) [116,117,118,119]. NRT1.1 is a dual-affinity nitrate transporter protein in *Arabidopsis* that acts as a plasma membrane nitrate receptor to sense external nitrate signals [120]. It has been reported that NRT1.1 relies on NO_3_^−^ for Na^+^ transport (Table 1). In the presence of nitrate, Na^+^ accumulation in the *nrt1.1* mutant was significantly lower than that of wild type [121]. However, when NH_4_^+^ was the only nitrogen source, the *nrt1.1* mutant significantly reduced the absorption of Cl^−^ and eliminated the salt-sensitive reaction caused by NH_4_^+^ after stress treatment with 25 mM NaCl [122]. ANR1 also responds to abiotic stress, and overexpression of *ANR1* produces a salt-sensitive phenotype (Table 1) [123]. The *NLP* gene family encodes core transcription factors that regulate nitrate signaling in plants [124]. In *Arabidopsis*, nine members of this family have conserved phosphorylation sites in the N terminus that are critical for responding to nitrate signaling [125,126]. *NLP8*, a nitrate-activated transcription factor, was found to activate the expression of the ABA catabolic gene *CYP707A2* in *Arabidopsis* and is essential for nitrate-induced seed germination (Table 1) [127]. The NR pathway is the main pathway for NO production [128]. Nitrate release of NO under NR can promote seed germination. In NaCl-stressed *Arabidopsis* seeds when exogenously treated with KNO_3_, nitrate serves as a nitrogen source and counteracts NaCl inhibition of seed germination by upregulating the expression of the *GA20ox1* gene encoding GA synthase. The *SPATULA* (*SPT*) gene encodes a bHLH transcription factor, and the *CYP707A2* gene encodes an ABA catabolic enzyme [114]. During the germination of cucumber seeds treated with 50 mM NaCl, NO-dependent NR activity also significantly improved salt tolerance [129]. Rice with defective *OsNR* gene function exhibits a phenotype that is sensitive to salt stress and shows delayed germination (Table 1). During rice seed germination, the nitrate-responsive transcription factor *NLP2* promotes the expression of NO-activated ABA catabolic genes *ABA8ox1* and *ABAox2* through the NR pathway, thereby improving plant salt tolerance [130]. NLP7 and TCP20 are major regulators of nitrate response genes in *Arabidopsis* roots and regulate the primary nitrate response [117,131], and both NLP6 and NLP7 strongly interact with TCP20 to influence cell proliferation factors that regulate the cell cycle [132]. Recently, deletion of the *NLP7* function has been shown to lead to a significant reduction in the transcript level of the ABA biosynthesis gene *NCED3*, which in turn improves salt tolerance in plants (Table 1) [133]. However, the interaction of other *NLP* family transcription factors with nitrate signaling response genes during salt stress-inhibited seed germination still needs to be further explored. The interaction of NO and ethylene also reduced the inhibitory effect of salt stress on seed germination [134]. It was found that the NO-dependent ethylene signaling factor *EIN3* can resist high salt stress to promote seed germination (Table 1) [135]. Additionally, recent studies have found that RNS can promote embryo germination of *S. pohuashanensis* under NaHCO_3_ stress mediated by exogenous ethylene [136].

#### 2.4.2. Mechanism of Nitric Oxide Action on Seed Germination under Drought Stress

NO can significantly increase the tolerance of plants to drought stress and improve seed viability and crop yield [137]. For example, the exogenous NO donors SNP and GSNO induced the transcription of *Oryza sativa* plasma membrane channel protein (*OsPIP1;3*), which in turn induced seed germination under water stress conditions (Table 1) [22]. Transformation of the unicellular marine alga *Ostreococcus tauri* NOS gene (*OtNOS*) into *Arabidopsis* significantly elevated NO accumulation and enhanced seed tolerance to salt stress [138].

ABA, an important signaling molecule for water stress, regulates the sensitivity of plants to ABA under osmotic stress by interacting with *GAP1*, the gene encoding the NAC family transcription factor *ANAC089* (Table 1) [139]. Further, inhibition of ABA synthesis under abiotic stress by *ANAC089* through regulation of NO homeostasis during seed development has been suggested [139]. Moreover, ABA-induced stomatal closure leads to a reduction in transpiration rate; NO mediation of stomatal development under drought stress also affects seed germination. Recent studies found that exogenous NO treatment mediated ABA synthesis in guard cells or induced quickly activating Anion Channel 1/Aluminum-activated Malate Transporter 12 (QUAC1/ALMT12) anion channels leading to stomatal closure and enhanced drought tolerance in plants [140]. In addition to exogenous NO affecting ABA synthesis, the response to the ABA signaling pathway via the NR pathway or NOA1-dependent NO production is important [141]. NO synthesis gene mutant *nia1nia2* results in lower than 1% bioactivity of NR and reduced accumulation of NO in guard cells [142]. Further, although *NIA1* and *NIA2* are functionally redundant, only *NIA1* is required for NO production in response to ABA [143]. It has been reported that *Atnoa1* mutants reduced NO response to ABA in root tips and guard cells [103]. Furthermore, the reduced NO content in *Atnoa1* mutants was associated with protein synthesis in chloroplasts [144]. In addition, *nia1nia2noa1-2* plants had reduced seed germination potential and were overly sensitive to ABA [137]. Hence, the interaction between NR/NIA and AtNOA1-mediated NO-ABA in guard cell signaling deserves considerable attention in future investigations into the molecular mechanisms of seed germination.

NO-dependent post-translational modifications also play an important role in the regulation of seed germination during abiotic stress [77]. NO inhibits the activity of SNF1-associated protein kinase (SnRK2.6/OST1) through GSNO, thereby inhibiting ABA signaling in guard cells (Table 1) [75]. Furthermore, dried seeds induced the expression of ABA-induced growth inhibitor ABI5; however, NO and ABA had antagonistic effects on ABI5 expression. Large amounts of NO are produced at the early stage of seed imbibition, which induces S-nitrosylation of ABI5 at Cys153 and promotes seed germination through proteasomal degradation [77]. NO not only affects seed germination through S-nitrosylation but also regulates the degradation of ethylene response factor ERFVII, a substrate of E3 ubiquitin ligase PROTEOLYSIS6 (PRT6), which suppresses ABI5 expression through ubiquitination modifications. *Arabidopsis prt6-1nlp7-1* seeds were found to be hypersensitive to ABA, suggesting that NLP7 and PRT6 interactions enhanced seed tolerance to ABA (Table 1) [76,145,146].

#### 2.4.3. Involvement of Nitric Oxide in the Regulation of Seed Germination by Ambient Temperature

Seed germination is often related to temperature, and low temperature stratification plays a positive role in the breaking dormancy of most seeds. Many studies have shown that RNS, ROS, and HCN can induce the breaking of deep dormancy during seed development. For example, SNP and cold stratification together significantly increased the germination rate of Empress Tree seeds. NO scavenger cPTIO significantly inhibited the rapid development of apple embryos pretreated with NO and HCN into seedlings [47,85]. Subsequently, it was found that NO or HCN induced transient accumulation of ROS during the enhancement of germination in apple embryos and sunflower (*Helianthus annuus*) embryos, which was associated with the AsA-GSH system [147,148,149]. The GSH system of seeds stratified at low temperatures for 40 d showed significantly higher glutathione peroxidase (GPX) and glutathione reductase (GR) activities, which in turn promoted seed germination [150]. GSNO is an important source of stable release of NO [151,152]. GSNOR and GR enzymes can break seed dormancy by regulating NO levels during low temperature stratification-mediated germination of apple embryos. Reduction of GSNO by GSNOR leads to the formation of oxidized forms of GSH and NH_2_OH, which in turn eliminates GSNO to maintain cellular RNS levels and prevent excessive accumulation of NO. GSSG is further reduced to GSH by the action of GR, which promotes seed germination [151,153,154].

However, ambient temperatures that are too low or too high often cause adverse effects on plants, resulting in suppressed seed germination and inhibited plant growth. A recent study found that irrigation of 89.4 mg/L SNP on two rapeseed (*Brassica napus* L.) varieties (ZY15 and HY49) significantly improved the resistance of rapeseeds to low temperature and drought stress, thus increasing the seed germination rate [155]. Tomato growth is inhibited at temperatures below 25 °C and growth nearly ceases below 6 °C. In tomato seeds, SNP treatment at 10 °C significantly increased amylase activity and soluble sugar content and improved low temperature tolerance [156]. In addition, KNO_3_ has an important effect on seed germination of *Sorbus pohuashanensis*. For seeds in long-term cold storage, incubation at variable temperatures (5 °C followed by 25 °C followed by 5 °C) after pretreatment with KNO_3_ significantly increased the germination rate compared to seeds incubated at 5 °C or 25 °C followed by 5 °C [157]. However, recent studies have found that methylglyoxal (MG) can promote maize germination by reducing endogenous NO levels. This finding suggests that, in addition to the degradation of ABA, NO may induce other signaling pathways to enhance maize germination at low temperatures (13 °C) [158].

High temperatures also reduced seed germination, while 1 mM SNP significantly increased the pod number and seed numbers in lentils (*Lens culinaris* Medik.) exposed to high temperature stress of 32 °C [159]. High temperature stress also led to a decrease in GSNO reductase (GSNOR1) activity preventing the degradation of the NO donor GSNO, which resulted in the accumulation of more RNS and reduced seed germination rate. Genetic analysis showed that overexpression of *GSNOR1* increased the nitrosation modification level of ABI5 and induced the degradation of downstream ABI5 protein increasing seed germination (Table 1) [160]. These studies suggest directions for future research on different types of gene modification to improve plant heat tolerance.

**Table 1 ijms-24-09052-t001:** Effects of NO-regulated target genes on seed germination under abiotic stress.

Stress Types	Gene	Species	Function Description	References
Salt Stress	*OsNOA1*	*Oryza sativa*	Overexpression of *OsNOA1* gene improved plant salt tolerance by reducing Na^+^/K^+^ ratio in the mutant.	[104]
	*GmNOS2* *GmNR1*	*Glycine max*	The Na^+^/K^+^ ratio was decreased and the balance of ABA, GA and IAA hormones was maintained, improving the salt tolerance index of soybean.	[106]
	*SlGR*	*Solanum lycopersicum*	Overexpression of *SlGR* significantly increased the activity of antioxidant enzymes and reduced the oxidative damage of tobacco seeds.	[111]
	*NRT1.1*	*Arabidopsis* *thaliana*	Na^+^ accumulation was promoted after NO_3_^-^ treatment, and Cl^-^ accumulation was promoted after NH_4_^+^ treatment.	[121,122]
	*ANR1*	*Arabidopsis* *thaliana*	Overexpression of *ANR1* produces a salt-sensitive phenotype and inhibited seed germination	[123]
	*NLP8*	*Arabidopsis* *thaliana*	Activate ABA catabolic gene *CYP707A2* to promote germination.	[127]
	*NLP2*	*Arabidopsis* *thaliana*	The NLP2-NR pathway activates the expression of ABA catabolic genes *ABA8ox1* and *ABAox2*.	[130]
	*NLP7*	*Arabidopsis* *thaliana*	Increasing the transcription level of ABA biosynthesis gene *NCED3* to inhibit seed germination.	[133]
	*NIA1,2*	*Arabidopsis* *thaliana*	It can resist salt stress by promoting EIN3 expression and transcription of downstream ethylene response genes.	[135]
Drought stress	*OsPIP1;3*	*Oryza sativa*	Overexpression of *OsPIP1;3* induced germination under water stress conditions.	[22]
Drought stress	*OtNOS*	*Arabidopsis* *thaliana*	Overexpression of *OtNOS* elevated NO accumulation and enhanced seed tolerance to salt stress.	[138]
	*GAP1*	*Arabidopsis* *thaliana*	Functionally deficient mutants showed insensitivity to ABA.	[139]
	*SnRK2.6*	*Arabidopsis* *thaliana*	NO inhibits the activity of SNF1-associated protein kinase (SnRK2.6) through GSNO, thereby inhibiting ABA signaling in guard cells.	[75]
	*PRT6-1* *NLP7-1*	*Arabidopsis* *thaliana*	The interaction between *NLP7* and *PRT6* enhanced seed tolerance to ABA.	[146]
Heat stress	*GSNOR1*	*Arabidopsis* *thaliana*	Overexpression of *GSNOR1* increased the nitrosation modification level of ABI5 and induced the degradation of downstream ABI5 protein increasing seed germination	[160]

## 3. Conclusions and Future Perspectives

Over the past 10 years, analysis of different types of RNS revealed that the seed germination rate could be significantly increased by gaseous NO or NO donor (SNP, SNAP, and GSNO) pretreatment. However, the NO synthesis and catabolism pathway remains a controversial topic. Although RNS and ROS can collectively affect seed germination, it has been reported that RNS and ROS were unstable and both induced and hindered seed germination. Studies showed that different concentrations of NO had different effects on seed germination. Low concentration of NO promoted seed germination, while high concentration of NO inhibited seed germination [18,20]. Moreover, exogenous NO interacted with phytohormone signaling pathways, such as ethylene synthesis, ABA catabolism, and GA synthesis, during early seed germination and increased seed germination by affecting the accumulation of ROS and the synthesis of enzymes related to the antioxidant system. However, excessive accumulation of H_2_O_2_ can also have a negative impact on plants. Although previously there have been extensive studies on the effect of NO on the physiological processes of seeds, whether the mechanisms of action on different phases of germination during the breaking of seed dormancy in different plants are similar or different remains an open question. In addition, the complex network of interactions between NO and different phytohormone signals deserves further exploration.

Under abiotic stress conditions, NO synthesis via the nitrate reduction (NR) pathway or NO synthase (NOS) has major effects on seed dormancy and germination. In the model plant *Arabidopsis*, the expression of downstream genes in the ABA and ethylene signaling pathways is regulated by the expression of *NLP* transcription factors in the NO_3_^−^ signaling pathway, which has an important function in improving plant tolerance to abiotic stress. This mechanism can also serve as a reference for the investigation of transcription factors during seed germination in non-model plants. Additionally, NO-dependent post-translational modifications (PTM), such as S-nitrosylation or ubiquitination, can regulate protein expression in response to stress. However, the effects of different types of PTM on NO-mediated seed germination require further investigation. With the continuous development of genome sequencing, the use of multi-omics to explore the key genes and gene interactions in the regulation by NO of seed dormancy and germination will provide a reference for the elucidation of the molecular mechanism of seed germination.

## Figures and Tables

**Figure 1 ijms-24-09052-f001:**
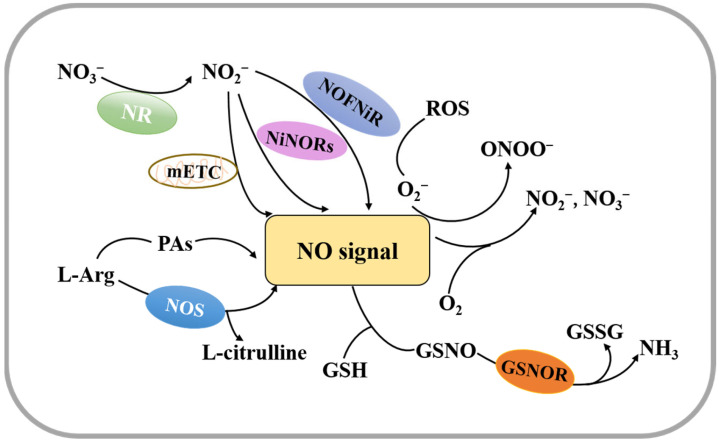
A model for the synthesis and decomposition of nitric oxide. NO is synthesized mainly through a nitrate/nitrite-dependent reduction pathway or through an NO synthase (NOS)-mediated oxidative pathway. NO is decomposed mainly by interacting with oxygen, ROS or GSH. NR; NO-forming nitrite reductase (NOFNiR); mitochondrial nitrite reductase electron transport chain (mETC); polyamines (PAs); GSNO reductase (GSNOR).

**Figure 2 ijms-24-09052-f002:**
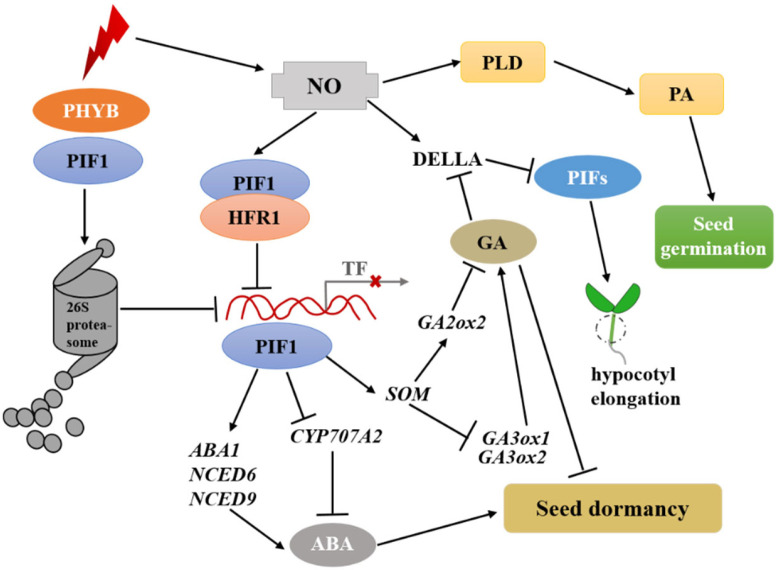
The effect of nitric oxide on seed germination under light. In the presence of red light, PHYB moves from the cytoplasm to the nucleus and promotes seed germination promoting expression of the 26S proteasome pathway leading to PIF1 degradation. NO accumulation under red light conditions not only inhibited PIF1 transcription but also enhanced the HFR1-PIF1 interaction, ultimately leading to a weakened inhibitory effect of PIF1 on seed germination. *SOM* as a direct target gene downstream of *PIF1*, can affect the expression levels of related genes in the ABA and GA signaling pathways in photochrome-dependent photomorphogenesis. Light-mediated an antagonism relationship between NO and GA. During seed germination, NO content and PLD activity are increased by light. The arrows and bars indicate positive and inhibitory effects, respectively. PHYTOCHROME-INTERACTION FACTOR 1 (PIF1); LONG HYPOCOTYL IN FAR-RED (HFR1); phospholipase D (PLD); phosphatidic acid (PA); ABA-DEFICIENT1 (*ABA1*); NINE-CIS-EPOXYCAROTENOID DEOXYGENASE6 (*NCED6*); GIBBERELLIN 2-OXIDASE2 (*GA2ox2*); *SOM* (*SOMNUS*).

**Figure 3 ijms-24-09052-f003:**
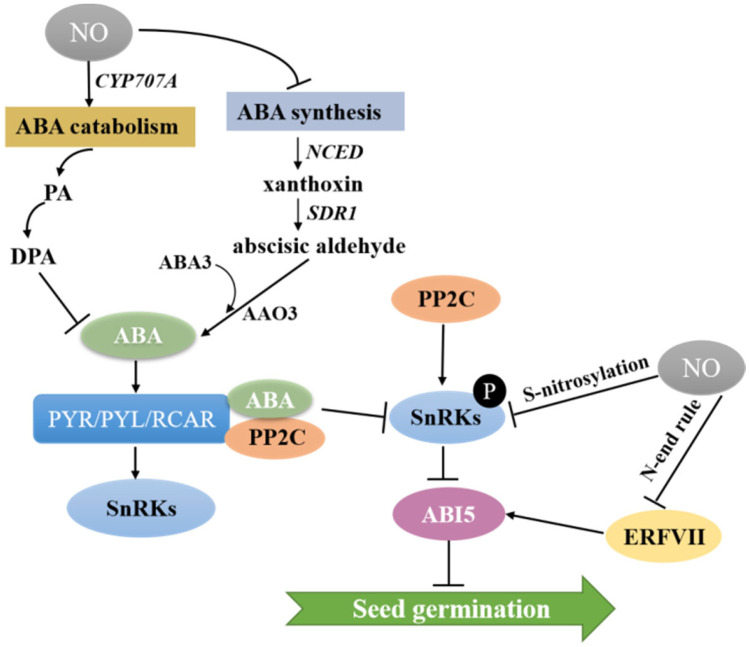
A model of NO-ABA regulation on seed germination. NO induces ABA catabolism and inhibits ABA synthesis: nine-cis-epoxycarotenoid dioxygenase (NCED); short-chain dehydrogenase reductase (SDR1), abscisic aldehyde oxidase 3 (AAO3); ABA DEFICIENT3 (ABA3); phaseic acid (PA); dihydrophaseic acid (DPA); type 2C protein phosphatases (PP2C); SNF1-related protein kinase (SnRKs); ABSCISIC ACID INSENSITIVE5 (ABI5); Group VII of the ethylene response factor (ERF/AP2) family (ERFVIIs).

**Figure 4 ijms-24-09052-f004:**
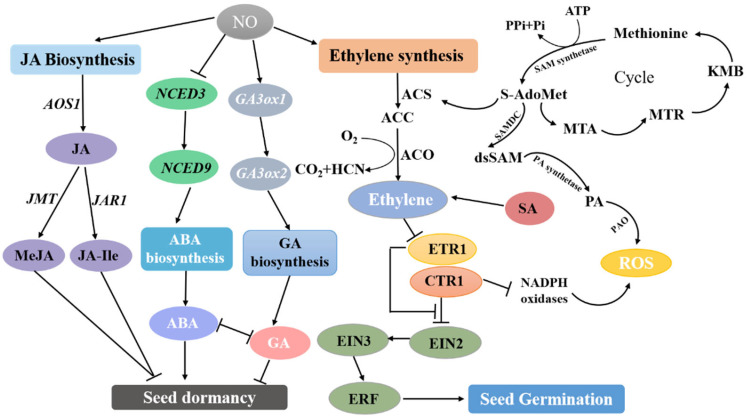
Interaction of NO with various phytohormones in the regulation of seed germination. NO promoted the GA, JA and ethylene synthesis pathways and inhibited the ABA synthesis pathway, respectively. methionine (Met); S-adenosyl-L-methionine synthetase (SAM synthetase); S-adenosyl-methionine (S-AdoMet); 1-aminocyclopropane-1-carboxylic acid (ACC); 5′-methylthioadenosine (MTA); S-adenosyl-L-methionine methylthioadenosine-lyase (ACS), ACC oxidase (ACO); hydrogen cyanide (HCN); ethylene resistant 1 (ETR1); constitutive triple response 1 (CTR1); ETHYLENE INSENSITIVE2 (EIN2); ethylene-responsive factor (ERF); JA methylester (MeJA) and JA-isoleucine (JA-Ile); Polyamines (PAs); SAM (dcSAM); SAM decarboxylase (SAMDC); PA oxidase (PAO).

## Data Availability

Not applicable.

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
