# Peer review of "Nitric Oxide Regulates Seed Germination by Integrating Multiple Signalling Pathways"

_ijms, 2023, doi:10.3390/ijms24109052_

Round 1

Reviewer 1 Report

The review of Zhang et al. is a thorough compilation of our current knowledge about the molecular plant mechanisms interconnected with Nitric oxide (NO), such as environmental conditions (such as light, temperature, and even electromagnetism), and hormones. I learned and enjoyed reading this review. Nonetheless, I believe its understanding will benefit by improving the following points.

Major points.

The article lacks uniformity. It reads as if the contributions from different people were just concatenated in this article. This becomes evident as how lengthy was the last part (abiotic responses) vs how much faster and dynamic was the first part (hormones and ROS). Also, there is no real uniformity in where the full name of a gene is introduced to the reader. Some genes are presented multiple times (e.g., NR, GR), while others are not. I will ask the authors to uniform this issue.

I would suggest the authors be more concise in the abiotic stress part of the review to improve its readability.

I would also suggest that the authors remove the abbreviations from the section titles since there are titles with concepts that have not been introduced before (e.g., 2.3.4… and PA on seeds…). Also, when the authors use NO-ET or NO-JA, it gives the false impression that there is a direct interaction between the hormones and NO.

Minor points.

L18. Hormones such as …

L82-83. It reads as the abbreviation for nitrate reductase, and the electron transport chain is mixed, please make this point clearer.

L120-122. Please re-write for clarity.

L140-141. It becomes a bit counterintuitive to follow this logic since, in the previous paragraph, the authors mentioned that PhyA is the receptor for far red light. And, as in described in this line, the light used for this reference was red light. Does this light contain as well far-red light or is PhyA also capable to respond to red light?

L154. PIF1 is been referred as a protein here, should not be in italics.

Figure 2. The author mentioned that the PIF1 would be translocated to the nucleus upon red light exposure. If in the model shown, the limited line refers to a plant cell, then the authors could include a scheme of which processes are taking place inside the cell nucleus. Otherwise, remove the border of the figure, as other figures do not have it anyway. Also, several abbreviations are absent in the figure legend (PA, ABA1, etc).

L224. NCED abbreviation is already mentioned.

L230. Please keep it consistent and change it to Arabidopsis.

L251. Unclear what is NO-ABA referred to. A node, interactions?

L254-255. This description is vague. What is being reduced in the N-end pathway? The oxidized cysteine after NO increase? Please be precise.

L285. Is SA = Salicylic acid?  If so, please specify.

L309. MV is a ROS-inducing herbicide.

L359. Not clear what Spm and Spd stand for

Table 1. Please change to Arabidopsis thaliana for consistency.

L379-381 and L120-122 needs re-writing for clarity

Author Response

Dear Reviewer :

We are very grateful to you for taking the time to read and review our paper again. We find that your comments play a very important role in improving the quality of our papers. We have carefully revised the paper in light of your comments, and please find our response to the comments made below. We marked the modified part of the manuscript in yellow.

Thank you for considering our revised manuscript!

Point 1: Major points.

The article lacks uniformity. It reads as if the contributions from different people were just concatenated in this article. This becomes evident as how lengthy was the last part (abiotic responses) vs how much faster and dynamic was the first part (hormones and ROS). Also, there is no real uniformity in where the full name of a gene is introduced to the reader. Some genes are presented multiple times (e.g., NR, GR), while others are not. I will ask the authors to uniform this issue.

Response 1: Thank you very much for your suggestion. We read the manuscript carefully and accepted your suggestions. We have modified the last part (abiotic responses) of the manuscript in yellow. In addition, We made uniformity for multiple genes. It has marked the modified part of the manuscript in yellow .

Point 2: I would suggest the authors be more concise in the abiotic stress part of the review to improve its readability.

Response 2: Thank you very much for your suggestion. We have carefully revised the paper and made the abiotic stress part of the review more concise. It has marked the modified part of the manuscript in yellow .

Point 3: I would also suggest that the authors remove the abbreviations from the section titles since there are titles with concepts that have not been introduced before (e.g., 2.3.4… and PA on seeds…). Also, when the authors use NO-ET or NO-JA, it gives the false impression that there is a direct interaction between the hormones and NO.

Response 3: Thank you very much for your suggestion. We read the manuscript carefully and accepted your suggestions. It has marked the modified part of the manuscript in yellow .

Point 4: Minor points.L18. Hormones such as …

Response 4: Thank you very much for your suggestion. We read the manuscript carefully and (Hormones such as) is added in line 17. It has marked the modified part of the manuscript in yellow.

Point 5: L82-83. It reads as the abbreviation for nitrate reductase, and the electron transport chain is mixed, please make this point clearer.

Response 5: Thank you very much for your suggestion. the mETC‐dependent reductions of nitrite to NO is mainly through electron transfer between complex III and complex IV. We read the manuscript carefully and modified the part of the manuscript. For details please see the text (2.1 in line 88).

Point 6: L120-122. Please re-write for clarity.

Response 6: Thank you very much for your suggestion. We read the manuscript carefully and modified the part of the manuscript. For details please see the text (2.2 in line 129).

Point 7: L140-141. It becomes a bit counterintuitive to follow this logic since, in the previous paragraph, the authors mentioned that PhyA is the receptor for far red light. And, as in described in this line, the light used for this reference was red light. Does this light contain as well far-red light or is PhyA also capable to respond to red light?

Response 7: In vitro, phyA and phyB have the similar spectral characteristics. The phytochromes mainly sense red light (600-700nm) and far-red light (750nm). KNO3 response to both far-red and red light during seed germination.

Point 8: L154. PIF1 is been referred as a protein here, should not be in italics.

Response 8: Thank you very much for your suggestion. We read the manuscript carefully and modified PIF1 in line 162.

Point 9: Figure 2. The author mentioned that the PIF1 would be translocated to the nucleus upon red light exposure. If in the model shown, the limited line refers to a plant cell, then the authors could include a scheme of which processes are taking place inside the cell nucleus. Otherwise, remove the border of the figure, as other figures do not have it anyway. Also, several abbreviations are absent in the figure legend (PA, ABA1, et)

Response 9: Thank you very much for your suggestion. We read the manuscript carefully and modified the figure and added several abbreviations.

Point 10: L224. NCED abbreviation is already mentioned.

Response 10: Thank you very much for your suggestion. We read the manuscript carefully and modified the NCED in line 234.

Point 11: L230. Please keep it consistent and change it to Arabidopsis.

Response 11: Thank you very much for your suggestion. We read the manuscript carefully and changed it to Arabidopsis (in line 239).

Point 12: L251. Unclear what is NO-ABA referred to. A node, interactions?

Response 12: Thank you very much for your suggestion. We read the manuscript carefully and modified the part of the manuscript (line 260).

Point 13: L254-255. This description is vague. What is being reduced in the N-end pathway? The oxidized cysteine after NO increase? Please be precise.

Response 13: Thank you very much for your suggestion. We read the manuscript carefully and modified the part of the manuscript (line 264-268).

Point 14:L285. Is SA = Salicylic acid?  If so, please specify. L309. MV is a ROS-inducing herbicide. ROS-generating compound

Response 14: Thank you very much for your suggestion. We read the manuscript carefully and added Salicylic acid(line 297) and ROS-generating compound (line 321)

Point 15:L359. Not clear what Spm and Spd stand for

Response 15: Thank you very much for your suggestion. We read the manuscript carefully and modified the part of the manuscript (line 372).

Point 16:Table 1. Please change to Arabidopsis thaliana for consistency.

Response 16: Thank you very much for your suggestion. We read the manuscript carefully and modified the Table1.

Point 17:Comments on the Quality of English Language

L379-381 and L120-122 needs re-writing for clarity

Response 17: Thank you very much for your suggestion. We read the manuscript carefully and re-writing for clarity. For details please see the text (line 129; line 393).

Reviewer 2 Report

Dear Authors,

The article is very interesting, especially since due to climate change, plants react differently to environmental conditions.

Therefore, research on seed germination is most needed.

Please consider changing the subject. Firstly, add in the name that THIS IS A REVIEW ARTICLE, and secondly, adjust the title to the content contained in the review. After all, the article is not only about the effect of nitric oxide on seed germination.

In lines 9 and 10 there are errors in the proper preparation of the work according to the requirements of the Journal.

The title of chapter 2 must necessarily be changed and refer to the content contained therein.

In the title of section 2.3.4. the abbreviation PAs should be explained (write the full name of Polyamines).

Or give the name of these compounds in the Summary.

Author Response

Dear Reviewer:

We are very grateful to you for taking time to read and review our paper again. We find that your comments play a very important role in improving the quality of our paper. We have carefully revised the paper in light of your comments, and please find our response to the comments made below. We marked the modified part of the manuscript in red.

Thank you for considering our revised manuscript!

Point 1: Please consider changing the subject. Firstly, add in the name that THIS IS A REVIEW ARTICLE, and secondly, adjust the title to the content contained in the review. After all, the article is not only about the effect of nitric oxide on seed germination.

Response 1: Thank you for the title suggested. The precedent version of the title has been replaced with A review of NO integrating multiple signaling pathways to regulate seed germination. It has marked the modified part of the manuscript in red.

Point 2: In lines 9 and 10 there are errors in the proper preparation of the work according to the requirements of the Journal.

Response 2: Thank you very much for your suggestion. We have carefully revised the paper in lines 9 according to the requirements of the Journal. It has marked the modified part of the manuscript in red.

Point 3: The title of chapter 2 must necessarily be changed and refer to the content contained therein.

Response 3: Thank you very much for your suggestion. We read the manuscript carefully and accepted your suggestions. For detail please see the text (2.2 in line 120; 2.3 in line 219). It has marked the modified part of the manuscript in red.

Point 4: In the title of section 2.3.4. the abbreviation PAs should be explained (write the full name of Polyamines). Or give the name of these compounds in the Summary.

Response 4: Thank you very much for your suggestion. We read the manuscript carefully and the full name of Polyamines is added in line 359. It has marked the modified part of the manuscript in red.

Reviewer 3 Report

The manuscript ijms-2375928 is a comprehensive review summarizes the role of NO together with other factors in regulating seed germination. The review is well written and organized. Although similar reviews are available, the current review is up to date and differently organized.

Some minor edits include the electric charge in chemical compounds which should be superscript.

It might be useful to start with seed dormancy vs seed germination then linking this to NO.

English language is fine.

Author Response

Dear Reviewer:

We are very grateful to you for taking time to read and review our paper again. We find that your comments play a very important role in improving the quality of our paper. We have carefully considered the suggestion of Reviewer and make some changes, and please find our response to the comments made below. We marked the modified part of the manuscript in purple.

Thank you for considering our revised manuscript!

Point 1: Some minor edits include the electric charge in chemical compounds which should be superscript.

Response 1: Thank you very much for your suggestion. We have carefully checked the manuscript and revised the superscript of the electric charge in chemical compounds. For details please see the text (page3, 7, 10, 11, 14). It has marked the modified part of the manuscript in purple.

Point 2: It might be useful to start with seed dormancy vs seed germination then linking this to NO.

Response 2: Thank you very much for your suggestion. We have carefully read the manuscript carefully and revised it according to your comments. Seed dormancy vs seed germination is a complex process. This process is affected by internal genetic and external environmental factors as well as available nitrogen sources. Light, temperature, nitrogenous compounds as signal inputs for seed dormancy and germination, activating changes in the levels of internal hormones such as ABA and GA [12]. However, the main form of nitrogen source is NO. It can promote seed germination by regulating ABA metabolism and GA synthesis pathways. We therefore linked seed dormancy vs seed germination to NO. For details of the changes, please see the text (Line 48-60, Page 2). It has marked the modified part of the manuscript in purple.